# Loop Unrolled Shallow Equilibrium Regularizer (LUSER) - A Memory-Efficient Inverse Problem Solver

## Abstract

In inverse problems we aim to reconstruct some underlying signal of interest from potentially corrupted and often ill-posed measurements. Classical optimization-based techniques proceed by optimizing a data consistency metric together with a regularizer. Current state-of-the-art machine learning approaches draw inspiration from such techniques by unrolling the iterative updates for an optimization-based solver and then learning a regularizer from data. This *loop unrolling* (LU) method has shown tremendous success, but often requires a deep model for the best performance leading to high memory costs during training. Thus, to address the balance between computation cost and network expressiveness, we propose an LU algorithm with shallow equilibrium regularizers (LUSER). These implicit models are as expressive as deeper convolutional networks, but far more memory efficient during training. The proposed method is evaluated on image deblurring, computed tomography (CT), as well as single-coil Magnetic Resonance Imaging (MRI) tasks and shows similar, or even better, performance while requiring up to $8\times$ less computational resources during training when compared against a more typical LU architecture with feedforward convolutional regularizers.

## 1 Introduction

In an inverse problems we face the task of reconstructing some data or parameters of an unknown signal from indirect observations. The forward process, or the mapping from the data to observations, is typically well known, but ill-posed or non-invertible. More formally, we consider the task of recovering some underlying signal $\boldsymbol{x}$ from measurements $\boldsymbol{y}$ taken via some forward operator $\boldsymbol{A}$ according to

$$\boldsymbol{y} = \boldsymbol{A}\boldsymbol{x} + \boldsymbol{\eta}, \tag{1}$$

where $\boldsymbol{\eta}$ represents noise. The forward operator can be nonlinear, but to simplify the notation, we illustrate the idea in linear form throughout this paper. A common approach to recover the signal is via an iterative method based on the least squares loss:

$$\widehat{\boldsymbol{x}} = \underset{\boldsymbol{x}}{\arg\min} \ \|\boldsymbol{y} - \boldsymbol{A}\boldsymbol{x}\|^2. \tag{2}$$

For many problems of interest, $\boldsymbol{A}$ is ill-posed and does not have full column rank. Thus, attempting to solve (2) does not yield a unique solution. To address this, we can extend (2) by including a regularizing term to bias the inversion towards solutions with favorable properties. Common examples of regularization include $\ell_2$, $\ell_1$, and Total Variation (TV). Each regularizer encourages certain properties on the estimated signal $\widehat{\boldsymbol{x}}$ (e.g., smoothness, sparsity, piece-wise constant, etc.) and is often chosen based on task-specific prior knowledge.

Recent works (Ongie et al., 2020) attempt to tackle inverse problems using more data-driven methods. Unlike typical supervised learning tasks that attempt to learn a mapping purely from examples, deep learning for inverse problems have access to the forward operator and thus should be able to guide the learning process for more accurate reconstructions. One popular approach to incorporating knowledge of the forward operator is termed *loop unrolling* (LU). These methods are heavily inspired by standard iterative inverse problem solvers, but rather than use a hand tuned regularizer, they instead learn the update with some parameterized model. They tend to have a fixed number

of iterations (typically around 5-10) due to computational constraints. Gilton et al. (2021) proposes an interesting alternative that takes advantage of *deep equilibrium* (DEQ) models (Bai et al., 2019; 2020; Fung et al., 2021; El Ghaoui et al., 2021) that we refer to as DEQ4IP. Equilibrium models are designed to recursively iterate on their input until a "fixed point" is found (i.e., the input no longer changes after passing through the model). They extend this principle to the LU method, choosing to iterate until convergence rather than for a "fixed budget".

**Our Contributions.** We propose an alternative novel architecture for solving inverse problems called *Loop Unrolled Shallow Equilibrium Regularizer* (LUSER). It incorporates knowledge of the forward model by adopting the principles of LU architectures while reducing its memory consumption by using a shallow (relative to existing feed-forward models) DEQ as the learned regularizer update. Unlike DEQ4IP that converts the entire LU architecture into a DEQ model, we only convert the learned regularizer at each stage. This has the advantage of simplifying the learning task for DEQ models, which can be unstable to train in practice. To our knowledge, this is the first use of multiple sequential DEQ models within a single architecture for solving inverse problems. Our proposed architecture (*i*) reduces the number of forward/adjoint operations compared to the work proposed by Gilton et al. (2021), and (*ii*) reduces the memory footprint during training without loss of expressiveness as demonstrated by our experiments. We empirically demonstrate better reconstruction across multiple tasks than LU alternatives with comparable number of parameters, with the ability to reduce computational memory costs during training by a factor of up to $8\times$.

The remainder of the paper is organized as follows. Section 2 reviews related works in solving inverse problems. Section 3 introduces the proposed LUSER, which we compare with other baseline methods in image deblurring, CT, and MRI tasks in Section 4. We conclude in Section 5 with a brief discussion.

## 2 RELATED WORK

### 2.1 LOOP UNROLLING

As noted above, a common approach to tackling an inverse problem is to cast it as an optimization problem consisting of the sum of a data consistency term and a regularization term

$$\min_x \ \|\boldsymbol{y} - \boldsymbol{A}\boldsymbol{x}\|_2^2 + \gamma\, r(\boldsymbol{x}), \tag{3}$$

where $r$ is a regularization function mapping from the domain of the parameters of interest to a real number and $\gamma \geq 0$ is a well-tuned coefficient. The regularization function is chosen for specific classes of signals to exploit any potential structure, e.g., $\|\boldsymbol{x}\|_2$ for smooth signals and $\|\boldsymbol{x}\|_0$ or $\|\boldsymbol{x}\|_1$ for sparse signals.

When $r$ is differentiable, the solution of (3) can be obtained in an iterative fashion via gradient descent. For some step size $\lambda$ at iteration $k = 1, 2, \ldots, K$, we apply the update:

$$\boldsymbol{x}_{k+1} = \boldsymbol{x}_k + \lambda \boldsymbol{A}^\top(\boldsymbol{y} - \boldsymbol{A}\boldsymbol{x}_k) - \lambda\gamma \nabla r(\boldsymbol{x}_k). \tag{4}$$

For non-differentiable $r$, the more generalized *proximal gradient* algorithm can be applied with the following update, where $\tau$ is a well-tuned hyperparameter related to the proximal operator:

$$\boldsymbol{x}_{k+1} = \mathrm{prox}_{\tau, r}(\boldsymbol{x}_k + \lambda \boldsymbol{A}^\top(\boldsymbol{y} - \boldsymbol{A}\boldsymbol{x}_k)). \tag{5}$$

The *loop unrolling* (LU) method performs the update in (4) or (5), but replaces $\lambda\gamma \nabla r$ or the proximal operator with a learned neural network instead. The overall architecture repeats the neural network based update for a pre-determined number of iterations, fixing the overall computational budget. Note that the network is only implicitly learning the regularizer. In practice, it is actually learning an update step, which can be thought of as de-noising or a projection onto the manifold of the data. LU is typically trained end-to-end. While end-to-end training is easier to perform and encourages faster convergence, it requires all intermediate activations to be stored in memory. Thus, the maximum number of iterations is always kept small compared to classical iterative inverse problem solvers.

Due to the limitation in memory, there is a trade-off between the depth of a LU and the richness of each regularization network. Intuitively, one can raise the network performance by increasing the number of loop unrolled iterations. For example, Gilton et al. (2021) extends the LU model to

potentially infinite number of iterations using an implicit network, and (Putzky & Welling, 2019) allows deeper unrolling iterations using invertible networks, while requiring recalculation of the intermediate results from the output in training phase. This approach can be computationally intensive for large-scale inverse problems or when the forward operator is nonlinear and computationally expensive to apply. For example, the forward operator may involve solving differential equations such as the wave equation for seismic wave propagation (Chapman, 2004) and the Lorenz equations for atmospheric modeling(Oestreicher, 2022).

Alternatively, one can design a richer regularization network. For example Fabian & Soltanolkotabi (2022) uses a transformer as the regularization network and achieves extremely competitive results in the fastMRI challenge (Zbontar et al., 2018), but requires multiple 24GB GPU for training, which is often impractical, especially for large systems. Our design strikes a balance in the expressiveness in regularization networks and memory efficiency during training. Our proposed work is an alternative method to achieve a rich regularization network without the additional computational memory costs during training.

## 2.2 DEEP EQUILIBRIUM MODELS FOR INVERSE PROBLEMS (DEQ4IP)

*Deep equilibrium* (DEQ) models introduce an alternative to traditional feed-forward networks (Bai et al., 2019; 2020; Fung et al., 2021; El Ghaoui et al., 2021). Rather than feed an input through a fixed (relatively small) number of layers, DEQ models solve for the "fixed-point" given some input. More formally, given a network $f_\theta$ and some input $\boldsymbol{x}^{(0)}$ and $\boldsymbol{y}$, we recursively apply the network via

$$\boldsymbol{x}^{(k+1)} = f_\theta(\boldsymbol{x}^{(k)}, \boldsymbol{y}), \tag{6}$$

until convergence.[1] In this instance, $\boldsymbol{y}$ acts as an input injection that determines the final output. This is termed the forward process. The weights $\theta$ of the model can be trained via implicit differentiation, removing the need to store all intermediate activations from recursively applying the network (Bai et al., 2019; Fung et al., 2021). In particular, we adopt the "Jacobian-free" backpropagation strategy outlined in Fung et al. (2021). This allows for deeper, more expressive models without the associated memory footprint to train such models.

Gilton et al. (2021) demonstrates an application of one such model, applying similar principles to a single iteration of an LU architecture. Such an idea is a natural extension as it allows the model to "iterate until convergence" rather than rely on a "fixed budget". More specifically, the model repeats (4) and (5) many times (in practice, usually around 50 iterations) until $\boldsymbol{x}_k$ converges. However, such a model can be unstable to train and often performs best with pre-training of the learned portion of the model (typically acting as a learned regularizer/de-noiser). It is also important to note is that such a model would have to apply the forward operator (and its adjoint) many times during the forward process. Although this can be accelerated to reduce the number of applications, it is still often signficiantly more than the number of applications for an LU equivalent which can be an issue if the forward operator is computationally expensive to apply.

## 2.3 ALTERNATIVE APPROACHES TO TACKLE MEMORY ISSUES

I-RIM (Putzky & Welling, 2019) is a deep invertible network that address the memory issue by recalculating the intermediate results from the output. However it is not ideal when forward model is computationally expensive. Gradient checkpointing (Sohoni et al., 2019) is another practical technique to reduce memory costs for deep neural networks. It saves intermediate activations of some checkpoint nodes, and recomputes the forward pass between two checkpoints for backpropagation. However, it is not an easy and efficient technique to implement for a weight-tied neural network.

## 3 METHODOLOGY

LU methods currently dominate the state-of-the-art approaches for solving inverse problems due to their stability when training, inclusion of the forward model, and their near instantaneous inference

---

[1]Note that, since our approach will ultimately use both methods, to aid in a clearer presentation we use subscript, i.e., $\boldsymbol{x}_k$, to denote the LU iterations, and superscript with parenthesis, i.e., $\boldsymbol{r}^{(i)}$, to denote the iterations in the deep equilibrium model.

times. However, there is a noticeable trade-off in terms of the memory requirements vs accuracy when training these models. Even for medium scale problems, some of the proposed architectures require multiple GPUs just to train, making this approach infeasible to use for much larger scale inverse problems. DEQ4IP offers an interesting alternative, drastically reducing the memory required during training and allowing the flexibility to adjust accuracy during reconstruction when performing inference. However, these models can potentially suffer when the forward/adjoint operators are computationally expensive, particularly since it takes more iterations to converge than a standard fixed LU method.

To address these concerns, we propose a novel architecture for solving inverse problems called *Loop Unrolled Shallow Equilibrium Regularizer* (LUSER). We adopt a LU approach to limit the number of forward/adjoint calls for particularly complex inverse problems while also drastically reducing the memory requirements for training, allowing us to scale up to much larger inverse problems (or require less GPU memory for existing problems). LUSER achieves this by adopting DEQ models as the trainable regularizer update in a standard LU architecture. The implicit DEQ models are smaller in size but just as expressive (Bai et al., 2019) as typical convolutional models used as the learned regularizer update allowing for an accurate reconstruction with less computational memory costs. Furthermore, learning a proximal update is a far simpler task compared to solving the inverse problem as a whole.

We adopt a "proximal gradient descent" styled LU architecture. The network takes in measurements $\boldsymbol{y}$ and some initial estimate $\boldsymbol{x}_0$. The architecture consists of $K$ stages, alternating between a gradient descent step $\boldsymbol{d}_k = \boldsymbol{x}_k + \lambda \boldsymbol{A}^\top(\boldsymbol{y} - \boldsymbol{A}\boldsymbol{x}_k)$ for $k = 1, 2, \dots, K$, followed by a feed-forward pass of the shallow equilibrium model acting as a proximal update block.

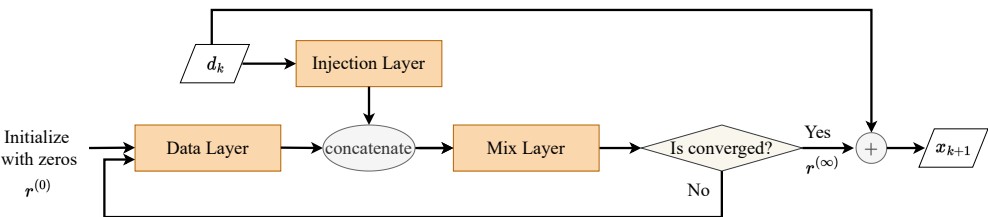

Figure 1: A proximal block in LUSER

Figure 1 illustrates a single block in LUSER as a learned proximal operator. The input $\boldsymbol{d}_k$ from the previous stage is processed only once by a set of input injection layers to avoid redundant computation. The input injection will determine the final fixed point output. The recursive portion of the proximal block consists of two sets of layers: the data layer and the mixing layer. The current estimate of the fixed point is first passed through the data layer before being concatenated with the input injection and processed by the mixing layer. This process repeats until a fixed point is found. In practice, convergence to a fixed point is achieved when the difference between two time steps $|\boldsymbol{r}^{(k+1)} - \boldsymbol{r}^{(k)}|$ is within a small $\epsilon$ or when the maximum number of iterations is reached. A more thorough summary can be found in Fung et al. (2021).

At each stage, the shallow equilibrium regularizer attempts to output its best estimate of the ground truth $\boldsymbol{x}^*$. We introduce a skip connection between the input $\boldsymbol{d}_k$ and the final output so that the regularizer is learning the residual $\boldsymbol{r}$ between the input $\boldsymbol{d}_k$ and the ground truth $\boldsymbol{x}^*$ instead, similar to the approach adopted by DnCNNs Ryu et al. (2019). When $\boldsymbol{x}_k$ converges to $\boldsymbol{x}^*$, $\boldsymbol{d}_k$ will also converge to $\boldsymbol{x}^*$ and we expect the residual to be closer and closer to the zero vector. Therefore, we initialize the input of data layer with a zero vector with the same dimensions as the input $\boldsymbol{d}_k$ in the hopes that fewer iterations will be needed at later stages as the current estimate $\boldsymbol{x}_k$ converges towards the ground truth. Let $\boldsymbol{r}^{(i)}$ denote the $i^{th}$ update of the residual, where $\boldsymbol{r}^{(0)} = \boldsymbol{0}$ and $\boldsymbol{r}^{(\infty)}$ be the fixed point solution of the residual. Let $\oplus$ denote the concatenation operator. The process in one loop unrolled block can be formulated as the following:

$$\boldsymbol{x}_{k+1} = \boldsymbol{d}_k + \mathrm{MixLayer}\left(\mathrm{DataLayer}(\boldsymbol{r}^{(\infty)}) \oplus \mathrm{InjectLayer}(\boldsymbol{d}_k)\right). \tag{7}$$

Although Figure 1 shows the simplest way of finding the fixed point, in practice acceleration or other fixed point solvers are applied to solve for the fixed point. Similar to Gilton et al. (2021), we

also apply Anderson acceleration (Walker & Ni, 2011) for all of our models when searching for the fixed point.

Using a fixed point solver based architecture for the regularization block allows LUSER to achieve a similar level of performance with a much shallower number of layers than traditional feedforward models. For the scope of this article, we restrict ourselves to a total of 5 layers to demonstrate memory savings, but initial experiments have shown that increasing the number of layers can yield higher performance.

We also explore two variants of LUSER, dubbed LUSER-SW and LUSER-DW. LUSER-SW refers to a shared-weight version of the proposed algorithm, where the proximal operator in all loop unrolled iterations are weight-tied and thus identical. Theoretically, the regularizer should be able to handle any input regardless of the iteration step. However, in practice, the distribution of intermediate reconstructions can be vastly different. Thus, training a single model to handle all these instances can be a daunting task, leading to poor generalization across the different stages. Since the total computational budget is fixed, one approach can be to use different weights (DW) for the learned proximal operator at each stage to handle the potentially different distributions. This will increase the number of parameters that need to be trained and stored, but since LUSER already has so few parameters to begin with, expanding to the different weight variant is still feasible. This paper will compare two variants of LUSER with other architectures in different tasks.

DEQ4IP relies on the learned proximal operator to have the same weights, thus we cannot include a different weight variant for comparison. On the other hand, the same principles can be applied to the LU variant of DEQ4IP. However, we aim to only compare models of similar number of parameters (or less in the case of LUSER-SW), and thus restrict our attentions to the shared weight variant of LU only.

## 4 EXPERIMENTS AND RESULTS

In this section, we compare our proposed networks to LU with DnCNN as proximal operator (LU-DnCNN) and DEQ4IP on three different tasks: image deblurring, computed tomography (CT), and single-coil accelerated Magnetic Resonance Imaging (MRI). The datasets we use are RGB CelebA (Liu et al., 2015), LoDoPaB-CT (Leuschner et al., 2021), and single-coil knee data from fastMRI (Zbontar et al., 2018) respectively. We also experiment with incorporating an auxiliary loss of including an MSE loss on intermediate reconstructions with the ground-truth instead of just the final output. This is done for the LU models (LU-DnCNN, LUSER).

### 4.1 EXPERIMENTAL SETUP

In the image deblurring task, the blurry image is obtained by applying a $(9 \times 9)$ Gaussian kernel with variance of $5$ to an image with additive white Gaussian noise with a standard deviation of 0.01. If the image is RGB, the same kernel is applied to all channels. For accelerated MRI tasks, measurements in k-space (or frequency domain) are often subsampled due to the cost in measurement. The goal of MRI reconstruction is to recover the underlying physical structure from subsampled noisy measurements. We simulate the forward operator $A$ with a 2-dimensional Fourier transform with randomly selected rows. We consider two common subsampling scenarios: $4\times$ and $8\times$ acceleration, or subsampling the columns in full measurement by a factor of 4 and 8 respectively. For the CT task, the forward operator is a Radon transform, and we uniformly select 200 out of 1000 angles in measurements. The adjoint of measurement $A^\top y$ is used as initialization for MRI and CT tasks, which brings the measurement back to the signal domain. However, in the deblurring task, since the measurement lies in a same domain as the underlying clean image, $y$ is used as the initial guess. Notice that some works use the filtered backprojection as the initialization for CT, such as (Khorashadizadeh et al., 2022), but we use the adjoint for the purpose of consistency.

We fix the budget of LU-DnCNN and LUSER to be a total of 8 iterations, while as we allow DEQ4IP to iterate until it reaches a fixed point. We use a DnCNN adopted from Ryu et al. (2019) with 17 convolutional layers with 64 channels, followed by BatchNorm and ReLU activations for the regularizer for LU-DnCNN and DEQ4IP. For the learned regularizer update in LUSER, we use 2 convolutional layers each for the input injection layer and data layer. The mixing layer contains 3 convolutional layers.

Table 1: Architecture of Proximal Network in LUSER

| | Layer Details |
|---|---|
| Injection Layer | SN(conv($C_{in}$:1, $C_{out}$:32, ks:3, pad:1)) + LeakyReLU
SN(conv($C_{in}$:32, $C_{out}$:32, ks:3, pad:1)) + LeakyReLU |
| Data Layer | SN(conv($C_{in}$:1, $C_{out}$:32, ks:3, pad:1)) GN + LeakyReLU
SN(conv($C_{in}$:1, $C_{out}$:32, ks:3, pad:1)) GN + LeakyReLU |
| Mix Layer | SN(conv($C_{in}$:64, $C_{out}$:64, ks:3, pad:1)) GN + LeakyReLU
SN(conv($C_{in}$:64, $C_{out}$:64, ks:3, pad:1)) GN + LeakyReLU
SN(conv($C_{in}$:64, $C_{out}$:1, ks:3, pad:1)) |

In order to stablize training for DEQ inspired models, we wrap all convolutional layers in DEQ4IP and LUSER with Spectral Norm (SN) (Miyato et al., 2018). We list more details for the learned proximal network for LUSER in Table 1 for the case when the input has a single channel. $C_{in}$ and $C_{out}$ denote the input and output channels, ks refers to the kernel size of a convolutional layer, pad denotes the padding in 2-dimension, and GN stands for GroupNorm.

We use two metrics to evaluate the quality of reconstruction: Peak Signal-to-Noise Ratio (PSNR) in dB and the Structural Similarity Index (SSIM). Note that although we use the same models as Gilton et al. (2021), we train our models from scratch and report lower values on the MRI task. We suspect that this is due to evaluating with a single channel only. When we include the imaginary channel (for a total of 2 channels), the metrics we recorded are more aligned with those reported in Gilton et al. (2021). All models are trained with a single RTX6000 24GB GPU.

## 4.2 RECONSTRUCTION QUALITY

Table 2 compares the average testing PSNR and SSIM. The different weight version of LUSER outperforms LU-DnCNN with a similar number of network parameters as specified in Table 3. LUSER-SW achieves similar level of performance in most tasks with only 5 layers, versus 17 layers in LU-DnCNN. DEQ4IP attains the best performance in image delurring task and CT, but LUSER achieves comparable quality. Training shared-weight architectures with auxiliary losses improves the reconstruction quality in most tasks. Note that we restricted ourselves to only 5 layers for LUSER-SW and LUSER-DW for all reported experiments to highlight the memory savings without degradation in performance. In an initial exploration, we noticed that increasing the number of layers can lead to boosts in performance. For example, in the CT task, LUSER-SW with 6 layers increases the PSNR and SSIM by 1.04 dB and 0.022 respectively.

Figure 2 shows representative reconstruction results. LUSER-DW achieves higher PSNR and SSIM in some cases, and attains better qualities in detailed structures, especially compared to LU-DnCNN. The areas with improvements are emphasized with red boxes in the ground truth images.

Table 2: Average PSNR and SSIM for test set, the best two performances are in bold.

| PSNR
SSIM | LU-DnCNN | | DEQ4IP | LUSER-SW | | LUSER-DW | |
|---|---|---|---|---|---|---|---|
| | Final loss | Aux loss | Final loss | Final loss | Aux loss | Final loss | Aux loss |
| Deblurring
$(3, 218, 178)$ | 29.93
0.862 | 30.39
0.871 | **31.57**
**0.895** | 30.30
0.869 | 30.65
0.878 | **31.40**
**0.891** | 31.15
0.888 |
| CT
$(1, 300, 300)$ | 30.59
0.844 | 31.59
0.859 | **32.19**
**0.871** | 28.82
0.801 | 28.04
0.797 | **31.83**
**0.860** | 31.66
0.859 |
| $4\times$ MRI
$(2, 320, 320)$ | 29.01
0.668 | 29.02
0.671 | 29.01
0.678 | 28.82
0.662 | 29.18
0.685 | **29.86**
**0.740** | **29.37**
**0.713** |
| $8\times$ MRI
$(2, 320, 320)$ | 27.50
0.576 | **27.65**
0.572 | 27.51
0.570 | 27.42
0.562 | 27.42
0.560 | **28.06**
**0.630** | 27.55
**0.596** |

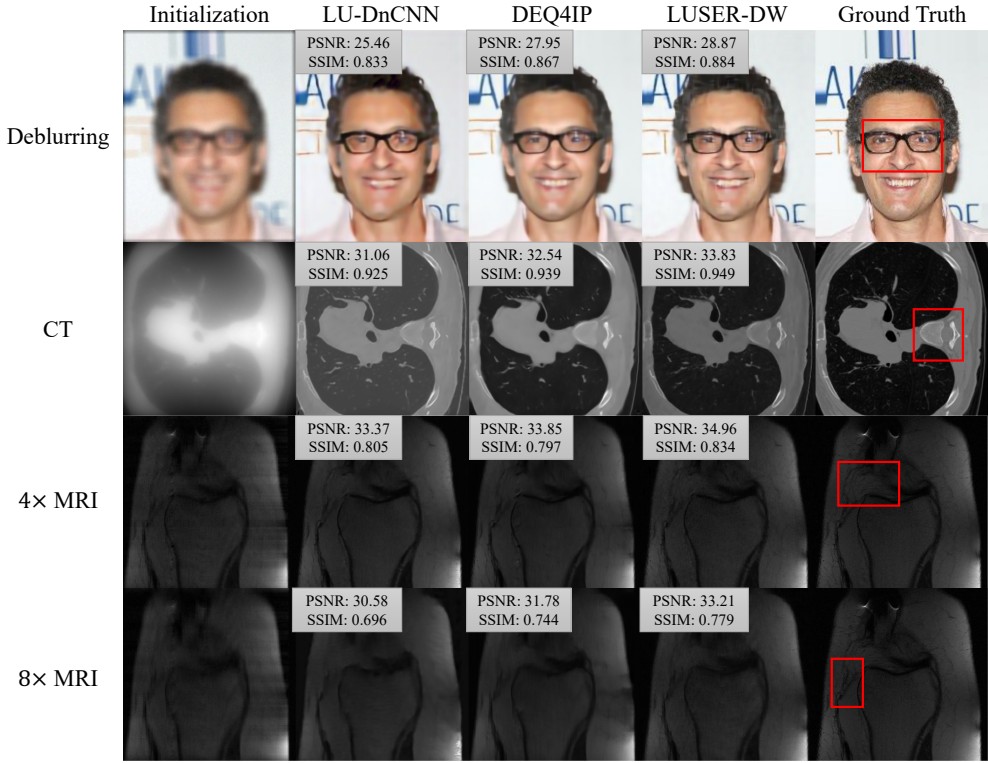

Figure 2: Representative reconstructions, where each row represents one task. The left-most column shows the initialization or input to the networks; the middle three columns show the reconstructions for LU-DnCNN, DEQ4IP and LUSER-DW; and the right-most column shows the underlying true images. Regions corresponding to qualitative improvements are emphasized in red boxes in the last column.

Table 3: Comparison of network sizes and maximum possible batch sizes during training. Entries with '-' denote that the architecture with a batch size of 1 cannot fit into a particular GPU RAM capacity.

| Maximum Batch Size | GPU RAM Capacity | **LU-DnCNN** (17 layers) | **DEQ4IP** (17 layers) | **LUSER-SW** (5 layers) | **LUSER-DW** (5 layers) |
|---|---|---|---|---|---|
| | #Params | 558,580 | 558,580 | 96,503 | 770,073 |
| Deblurring | 8 GB | 1 | 16 | 10 | 10 |
| $(3, 218, 178)$ | 10 GB | 1 | 16 | 14 | 14 |
| | 24 GB | 4 | 68 | 34 | 34 |
| | #Params | 556,033 | 556,033 | 93,954 | 751,625 |
| CT | 8 GB | - | 4 | 2 | 2 |
| $(1, 300, 300)$ | 10 GB | - | 6 | 4 | 4 |
| | 24 GB | 2 | 20 | 10 | 10 |
| | #Params | 557,185 | 557,185 | 95,107 | 760,849 |
| MRI | 8 GB | - | 4 | 4 | 4 |
| $(2, 320, 320)$ | 10 GB | 2 | 6 | 4 | 4 |
| | 24 GB | 4 | 16 | 12 | 12 |

### 4.3 MEMORY IN TRAINING

GPU capacity is a major bottleneck for training large-scale loop unrolled networks as discussed earlier. Table 3 compares the network sizes (number of parameters) as well as the maximum training batch sizes for three commonly seen GPU RAM capacities: 8 GB, 10 GB and 24 GB. Batch sizes are recorded with maximum even numbers, except when it is 1 for stochastic gradient descent. Notice that in MRI, because the Fourier transformation is implemented in tensor form, the minimum batch size it can take is 2. We use the batch size as a proxy for the memory requirements during training. Since DEQ4IP is an extension of LU-DnCNN, their networks are of the same size, but implicit DEQ models support larger batch sizes making DEQ4IP far more memory-efficient during training. The advantages of using DEQ models for LUSER and DEQ4IP are particularly highlighted in the case of limited memory (smaller GPUs). LU-DnCNN is unable to even train for the CT and MRI task with limited memory constraints, while LUSER and DEQ4IP can. This pattern is expected to repeat for more large scale tasks where standard LU architectures will be unable to train at all due to memory requirements. It is important to note that memory requirements depend more on the depth of the network than the number of parameters. For example, even though LUSER-DW and LUSER-SW have different numbers of parameters, they share the same architecture/depth and thus use roughly the same amount of memory during training.

### 4.4 INFERENCE TIME

The evaluation time of these models depend on the complexity of the forward model. For the experiments performed, taking a gradient step on average took 5.3e-5, 6.9e-4, and 6.6e-3 seconds for deblurring, MRI and CT respectively. In this regime, the time to evaluate the DEQ models dominates the inference time. However, as the forward model time increases, it has the potential to dominate.

In order to test this, we introduce artificial delays of 5e-2 and 5e-1 seconds to the CT forward model and evaluate the models on the test set. Because the performance does not significantly vary (as the trained models didn't change), we only report the run times in Figure 3. In these regimes, the choice of adopting a Loop Unrolled structure presents an advantage over DEQ4IP. In our experiments, DEQ4IP took approximately 30-50 iterations to reach a reasonable level of performance, leading to 30-50 calls of the forward model. The inference time is shown in the shaded region in green of Figure 3a, while as LUSER is shown in red for a max iteration of 10-30. It is interesting to note that as the forward model run time increases, so does the overlap between LUSER and DEQ4IP, with LUSER evaluating faster on average somewhere between 1e-2 and 1e-1 seconds.

The reason becomes more evident in Figure 3b which show cases the breakdown in timing for the solid line example from Figure 3a. The blue bar represents the approximate portion of time spent evaluating the forward model while as the green and red represent the learned model portion (as well as any overhead) associated with DEQ4IP and LUSER respectively. As the forward modeling time

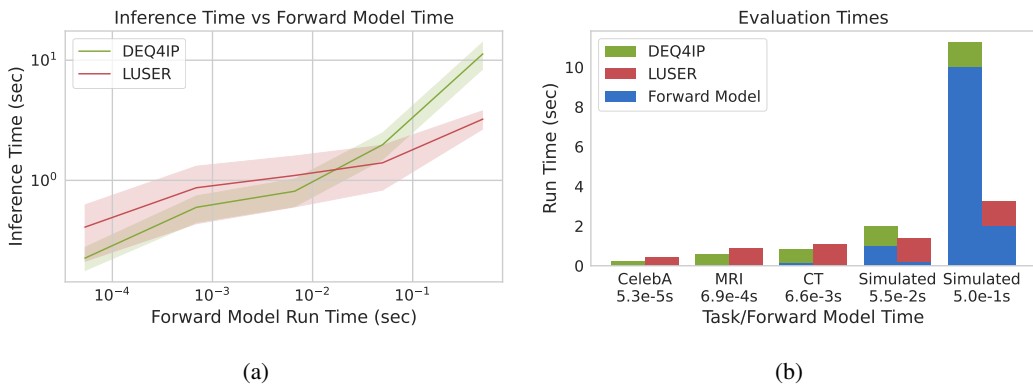

|       | (a) | (b) |

Figure 3: (a) Plot of inference time across a range of "max iterations" and (b) with approximate forward model time overlaid on total time.

becomes non-negligible, the evaluation time suffers accordingly with LUSER evaluating faster even for the 5e-2 second delay experiment.

## 4.5 EFFECT OF WEIGHT SHARING

As shown in Table 2, LUSER-DW has better performance than LUSER-SW with equivalent block structure, but has a larger number of parameters. We also explore the possibility of reusing multiple LUSER proximal blocks across loop unrolled iterations. For example, in MRI tasks, we repeat 4 different proximal blocks over 8 iterations. Let $g_i$ denote the $i^{th}$ proximal block where $i = \{1, 2, 3, 4\}$, and we form the LUSER network with the sequence of proximal blocks in order of $g_1, g_1, g_2, g_2, g_3, g_3, g_4, g_4$. Now, the number of parameters is only half of that in LUSER-DW, but maintains a similar level of performance. We denote this variant LUSER-PSW, which stands for partially shared-weight. Table 4 compares the average PSNR and SSIM of LUSER-SW, LUSER-PSW and LUSER-DW with the same block structure, which are trained using final loss only.

Table 4: Average PSNR and SSIM for LUSER with different weight-sharing schematics

|              |      | LUSER-SW | LUSER-PSW | LUSER-DW |
|--------------|------|----------|-----------|----------|
| $4\times$ MRI | PSNR | 28.82    | 28.85     | 29.86    |
|              | SSIM | 0.66     | 0.74      | 0.74     |
| $8\times$ MRI | PSNR | 27.42    | 28.04     | 28.06    |
|              | SSIM | 0.56     | 0.63      | 0.63     |

Table 5 summarizes various prorperties among the three architectures. In particular, we refer to the expressiveness of the network loosely as the performance relative to the depth. LUSER is able to achieve comparable performance to both DEQ4IP and LU with a much shallower network. We denote the rows related to timing with the asterisk to include the simulated regimes as well, though for negligible forward models, DEQ4IP can be faster than LUSER.

Table 5: Method Comparisons

|                       | LU    | DEQ4IP   | LUSER    |
|-----------------------|-------|----------|----------|
| Training Time*        | Fast  | Slow     | Moderate |
| Inference Time*       | Fast  | Slow     | Moderate |
| Network Size          | Large | Large    | Small    |
| Training Memory Usage | Large | Small    | Moderate |
| Expressiveness        | Low   | Moderate | High     |

## 5 CONCLUSION

Loop unrolling architectures with deep convolutional layers as the learned regularizer update are a popular approach for solving inverse problems. Although its variants achieve state-of-the-art results across a variety of tasks, LU algorithms incur a huge memory cost when training due to the requirement of saving all intermediate activations, sometimes even requiring multiple GPUs to train on complex tasks. DEQ4IP offers an interesting alternative via extending LU to infinitely many layers by finding a fixed point solution, but can be impractical when the forward/adjoint operators are nonlinear or larger in scale. To address the memory issue, we proposed two variants of loop unrolling architectures with deep equilibrium models as the learned regularizer updates, LUSER-SW and LUSER-DW. We verify the memory savings (by comparing batch sizes) relative to loop unrolling algorithms with a DnCNN model. Across all tasks, LUSER-DW outperforms LU-DnCNN with a similar number of network parameters, while reducing the memory requirements by a factor of 5 or more.

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
