# OpenReview forum: "Loop Unrolled Shallow Equilibrium Regularizer (LUSER) - A Memory-Efficient Inverse Problem Solver"
_ICLR.cc/2023/Conference — Submitted to ICLR 2023_

### Official Review · Reviewer_VVvv · 2022-10-23

**Confidence:** 4
**Correctness:** 3
**Technical Novelty And Significance:** 3
**Empirical Novelty And Significance:** Not applicable
**Recommendation:** 6

**Clarity, Quality, Novelty And Reproducibility:**

The paper is well written, there seems to be enough info to reproduce the results, their are substantial memory savings associated with the method (though other info is missing), and the proposed technique, while perhaps a natural evolution of existing ideas, is to my knowledge novel.

**Strength And Weaknesses:**

# Strengths
Solving large scale imaging inverse problems with unrolled network architectures is an important task and memory efficient training is one of the largest open problems in this field. This work represents a meaningful step in solving this problem.

The paper is generally well-written and easy to follow.

The proposed network architecture is well-motivated and simple -- this is a good thing.

# Weaknesses
While the paper is not claiming to be state-of-the-art, the comparisons in the paper are somewhat lacking and the unrolled DnCNN baseline is far from state-of-the-art. Even removing biases from the network may have significantly increased performance [A] so it's unclear if there are really any performance (rather than memory) benefits associated with the proposed method.

[A] Mohan, Sreyas, et al. "Robust And Interpretable Blind Image Denoising Via Bias-Free Convolutional Neural Networks." International Conference on Learning Representations. 2019.

The training procedure description states only that "The weights \theta of the model can be trained via implicit differentiation, removing the need to store all the intermediate activations". This could use a citation or additional explanation.

A number of statements lack evidence or citations. E.g., "The implicit DEQ models are smaller in size but just as expressive as typical convolutional models".

While Table 4 is a useful summary of the relative difference in training times, evaluation times, etc., the actual training times, evaluation times, etc. need to be in the paper. "Fast" vs "slow" is almost meaningless without context and this imprecision makes it nearly impossible to fully evaluation the submission -- the proposed method could have 1 day inference times.

In table 5, two comparisons isn't enough to show any trends. Would 7 layers further improve performance? When does performance plateau?

I would avoid stating LUSER provides a path forward for non-linear inverse problems. While this may be true, there's zero evidence for this in the paper.


**Summary Of The Paper:**

This work proposes a new unrolled network architecture for inverse problems. It's main contribution is to replace the large feed-forward architecture (e.g., DnCNN) used in typical unrolled architectures with a small recurrent architecture, known as a Deep Equilibrium (DEQ) model. The DEQ model iterates until it converges to a fixed point and only the gradients associated with this fixed point need to be stored, thus making the proposed architecture more efficient to train.

The proposed architecture is applied to simulated image deblurring, CT, and MRI where it performs incrementally better than a standard unrolled projected gradient descent baseline. The proposed architecture is far more memory efficient to train than the baseline, but actual training times are not provided.


**Summary Of The Review:**

The paper proposes a simple and novel technique for significantly reducing the memory costs associated with solving large scale imaging inverse problems with unrolled neural networks. While the current submission is missing important info (e.g., inference times), I remain mostly positive about this paper.

---

> ### Author Response · Authors · 2022-11-18
> **Response to Reviewer VVvv**
>
> We thank Reviewer VVvv for their thoughtful review. In particular, we appreciate that they found our proposed
> architecture simple and well-motivated. We respond to their highlighted issues below.
>
> 1. We appreciate the reviewer bringing [1] to our attention. We agree that some minor tweaks such as removing
> biases (although our implementation also does not include a bias) could lead to notably different performance.
> Our primary goal was indeed focused on memory reduction, with performance gains being a unexpected but
> pleasant secondary benefit. We believe this can be attributed to the expressiveness of DEQ models in general,
> though acknowledge the corresponding trade-offs (longer inference times).
> 2. With regards to training procedures, we have added the corresponding citation for increased clarity. In
> particular we use the jacobian-free approach outlined in [2].
> 3. We base our claim on the expressiveness of DEQs off of the works of [2], [3] who directly compare implicit
> models against traditional forward models and demonstrate comparable/better performance. However, we
> acknowledge that "expressive-ness" is not rigorously defined so we have walked back the claim in the paper.
> 4. We agree that timing is a crucial piece of information for evaluating our work. We have restructured our results
> section to incorporate this feedback as well as collected more timing information. We note that for the current
> tasks, there are certain settings (in particular the max number of iterations of the DEQ modules) that play
> a large role in the evaluation time. Our analysis shows that some settings lead to comparable run times for
> LUSER vs DEQIP, with DEQIP achieving slightly faster speeds, though nowhere near as fast as LU-DnCNN.
> However, as the complexities of forward models increase, we believe our choice of adopting a LU structure
> over the DEQIP will become an advantage leading to faster LUSER evaluations.
> To demonstrate this, we conducted an additional set of experiments with an artificially introduced time delay in
> the forward model, as demonstrated in Section 4.4 titled Inference time. In the regime of merely 0.05 seconds
> for a single forward pass, LUSER already begins to demonstrate some of its advantages, evaluating faster than
> DEQ4IP. With more limited settings (10 max iters for LUSER), we also evaluate faster than DEQ4IP for the
> non-simulated forward models. It is not unreasonable to encounter forward models as slow as 2 minutes [4],
> so we believe these trade-offs are worth considering for future experiments.
> 5. Our intention was to show that we focused primarily on memory savings with a highly restrictive shallow
> network that was still able to perform surprisingly well, but extending with additional layers can lead to
> increased performance. However, you are correct that trends cannot be drawn with just two data points.
> Unfortunately, we do not believe we have the time to extend the table, so have removed it from the final draft.
> 6. Statement has been removed at the reviewer’s suggestion.
>
> [1] S. Mohan, Z. Kadkhodaie, E. P. Simoncelli, and C. Fernandez-Granda, “Robust and interpretable blind image
> denoising via bias-free convolutional neural networks,” arXiv preprint arXiv:1906.05478, 2019.
> [2] S. W. Fung, H. Heaton, Q. Li, D. McKenzie, S. J. Osher, and W. Yin, “Fixed point networks: Implicit depth models
> with jacobian-free backprop,” 2021.
> [3] S. Bai, J. Z. Kolter, and V. Koltun, “Deep equilibrium models,” Advances in Neural Information Processing
> Systems, vol. 32, 2019.
> [4] M. Louboutin, “Modeling for inversion in exploration geophysics,” Ph.D. dissertation, Georgia Institute of
> Technology, 2020.

---

### Official Review · Reviewer_ko7X · 2022-10-25

**Confidence:** 2
**Clarity, Quality, Novelty And Reproducibility:** The manuscript is not clear. I am not…
**Correctness:** 2
**Technical Novelty And Significance:** 1
**Empirical Novelty And Significance:** 1
**Recommendation:** 3

**Strength And Weaknesses:**

*Weakness

1. The paper is not well written. The methodology section lacks enough technical details as well as explanations for readers to understand the proposed model. For example, Fig. 1 is not illustrative, and the caption does not provide any explanation of the figure. Due to the above weakness, I am not equipped with enough knowledge of the model to evaluate the contribution of the model and interpret the numerical results.

2. Please explain what is key difference between LUSER and prior models, such as DEQ4IP. I think DEQ4IP also only learns the regularizer network, isn't it? (Data-consistency layer is formed using the fixed forward model, while the regularizer layer is a neural network).

3. The resolution of Fig. 2. is too low to see the visual difference.

**Summary Of The Paper:**

The paper aims to propose a memory-efficient deep equilibrium model (DEQ) for solving inverse problems. However, after reading the paper, I am not sure if I have a clear understanding of what are the key contributions of the proposed LUSER model. It seems to me that the key difference between LUSER and prior DEQ models are:

1. LUSER uses pre-learned DEQ models in its architecture (*"...LUSER achieves this by adopting DEQ models as the learned
regularizer update in a standard LU architecture...."*)
2. Somehow a shallow network can be used so that LUSER only needs to update a small number of parameters so that the memory requirement can be reduced.

**Summary Of The Review:**

Not a good paper.

---

> ### Author Response · Authors · 2022-11-18
> **Response to Reviewer ko7X**
>
> We thank Reviewer ko7X for their thoughtful review. We provide some clarifying points with respect to their summary
> below:
> 1. The main distinction between DEQ4IP [1] and our work is where the idea of deep equilibrium models are
> applied. DEQ4IP converts the entire network including the gradient step into an equilibrium model, while
> as we only convert the learned (not pre-trained in our case) regularizers into equilibrium models (DEQ). We
> believe that learning a regularizer is a simpler task and comes with some interesting trade-offs compared to
> learning the entire optimization solution.
> 2. It’s important to clarify the usage of memory under the two contexts talked about in the paper. The first
> relates to the storage memory of the network and is closely tied to the number of parameters. The second
> reference of memory corresponds to memory used during training (typically the RAM of a GPU). Although
> storage/number of parameters play a role in memory usage during training, it is not the main contributor. We
> make this distinction because although our proposed models may have a similar number of parameters/storage
> memory, it does use significantly less training memory, which is the bigger bottleneck in training such models.
> This is typically heavily correlated with the depth of the network, hence our use of "Shallow Equilibrium
> Regularizers", a trade-off enabled by the usage of DEQs.
>
> We now respond to their highlighted issues below.
> 1. We have updated Section 3 "Methodology" in the hopes of providing more technical details. We refer the
> reviewer to paragraph 3-5 of the section for an explanation of the model. In particular, we adopt a very similar
> model to the one proposed by [2], but introduce a skip connection between the input injection and the output.
> 2. The baseline model is a loop unrolled architecture that performs a fixed number of alternating steps between
> gradient descent and a learned regularizer update. DEQ4IP uses the same regularizer update, but instead of
> only performing a fixed number of steps, uses ideas from the equilibrium model literature to "iterate until
> convergence"
> Alternatively, our proposed model stays within the loop unrolled framework with a fixed number of iterations
> and instead replaces the learned regularizer update model with one based on a deep equilibrium model.
> Although this seems similar to previous works that propose novel architectures, we believe the unique trade-
> offs introduced (lower training memory usage, scalability to slower forward models, etc.) by using DEQs
> principles merits discussion beyond the specific architecture used.
> 3. Figure 2 has been updated with increased size as well as metrics overlaid.
>
> [1] D. Gilton, G. Ongie, and R. Willett, “Deep equilibrium architectures for inverse problems in imaging,” IEEE
> Transactions on Computational Imaging, vol. 7, pp. 1123–1133, 2021.
> [2] S. W. Fung, H. Heaton, Q. Li, D. McKenzie, S. J. Osher, and W. Yin, “Fixed point networks: Implicit depth models
> with jacobian-free backprop,” 2021.

---

### Official Review · Reviewer_4Y5M · 2022-10-25

**Confidence:** 4
**Correctness:** 1
**Technical Novelty And Significance:** 2
**Empirical Novelty And Significance:** 2
**Recommendation:** 3

**Clarity, Quality, Novelty And Reproducibility:**

The paper is not well-written and contains lots of grammar errors. The novelty and experiments are limited.

**Strength And Weaknesses:**

Strengths:

1. The paper reduces the memory cost for loop unrolling by combining it with the deep equilibrium models.

Weaknesses:
1) The novelty of this paper is quite limited. Authors combine LU and DEQ, but results seem to be worse than DEQ. Author claims their approach to be better than LU and DEQ in several aspects such as stability. However, extensive experiments have not been conducted to backup these claims.

2) Authors claim for even medium scale problems there is a noticeable tradeoff for accuracy for LU models. Authors have focused on 2D data to show that their approach requires less memory and can deal with large scale complex inverse problems. However, for many applications involving complex operations such as MRI,  LU have achieved a state-of-the-art results (see fastMRI challenge, Muckley et. al, TMI 2021) for 2D datasets.  Thus, memory cost has not been a main challenge for processing 2D data. This issue arises when processing 3D datasets. However, authors have not included any experiments focusing on processing 3D datasets.

3) Authors mention that their approach can tackle stability issues seen in DEQ4IP.  However, there is no experiment backing up this claim.

4) Experimental results from Table 2 shows that, DEQ4IP outperforms LUSER. Luser can *quantitatively* achieve slightly better performance when parameters are not shared.  However, quantitative metrics may not be reflective of the performance.  In figure 2, MRI images for LUSER approach seem to be more blurry than DEQ4IP. Similarly, DEQ4IP achieves sharper results for CT. Thus, in practice, LUSER provides worse results than the counterpart approach. Also, why LUSER-SW is not included in figure 2?



**Summary Of The Paper:**

This paper combines algorithm unrolling, i.e. loop unrolling, with the deep equilibrium models for reduced memory and improved results.

**Summary Of The Review:**

The paper combines loop unrolling with deep equilibrium models. The experimental results show that the proposed approach does not achieve better results than DEQ. Its benefit over LU is also questionable as the paper does not show results for a large-scale problem such as processing 3D datasets.

---

> ### Author Response · Authors · 2022-11-18
> **Response to Reviewer 4Y5M**
>
> We thank Reviewer 4Y5M for their thoughtful review. We respond to their highlighted
> issues below
>
> 1\. While as DEQ4IP presents a very interesting methodology for solving inverse problems, it still has its own
> limitations and trade-offs. Our primary goal with presenting LUSER was to emphasize the memory savings
> without loss of expressiveness of the network compared to existing LU architectures. We acknowledge that
> from this perspective alone, DEQ4IP presents a far stronger case, achieving even more memory savings and
> similar if not better levels of performance than the baseline.
>
> However, as a secondary goal (although not present in our current experiments), we wish to tackle more
> expensive forward models. In this regime, where applying the forward model and its adjoint dominates the
> computational time relative to the DNNs, we believe LUSER provides a better trade-off. Experimentally,
> most of the DEQ4IP models required at least 30 iterations (and thus 30 applications of the forward model) to
> converge, while as LUSER only requires 8 while still achieving comparable results.
>
>   To highlight this scenario/regime where the trade-offs of LUSER provide an edge over DEQ4IP, we included a
> new set of simulated examples (with artificially introduced delays) of increasing forward model times. For the
> relatively modest example of a run time of 0.05 seconds, LUSER begins to outperform DEQ4IP in terms of
> speed. Some tasks have forward modeling times on the order of seconds, with some seismic tasks taking over
> 2 minutes [1].
>
> 2\. Although memory costs have not been a limiting factor for 2D, we opted to start with 2D as a more pragmatic
> ground to explore the ideas outlined in our work. We believe that the memory savings demonstrated by LUSER
> suggests a promising avenue to tackle 3D experiments in the future.
>
> Our primary focus was reducing the training memory footprint while minimizing drops in performance, which
> we believe the experiments in our work verify.
>
> 3\. Unfortunately, all our experiences with regards to stability during training are only anecdotal in nature. We
> have discussed possible extensions such as a test of sensitivity for convergence with respect to a hyperparameter
> sweep, but do not believe we would have the results in time for this conference. As such, we will walk back
> the claims in our introduction as, like you mentioned, we cannot support it with concrete evidence.
>
> However, we would like to note that we selected the forward models/data specifically because we knew that
> DEQ4IP was stable in such settings, but still ran into issues getting models to converge in our initial exploration.
> The original work also encountered similar issues and explicitly added checkpointing to previous states when
> loss spikes were detected. We encountered no such issues in training LUSER. It is worth noting that DEQs
> are still currently a fairly new architecture and not nearly as well understood as feedforward models and thus
> currently difficult to train in practice. However, our conjecture is that the task we assign the DEQ to train in
> LUSER (learning a regularizer update) is significantly easier than the one in DEQ4IP (solving the inverse
> problem), thus contributing to our anecdotal experience of more stable to train.
>
> 4\. We have included additional examples demonstrating instances where LUSER not only quantitatively but
> also qualitatively out-performs DEQ4IP. However, we believe this will vary case by case. Overall our main
> focus has been on memory savings with respect to LU architectures and removing potential computational
> bottlenecks with respect to DEQ4IP without reducing overall performance. We believe that our proposed
> model accomplishes this goal.
>
> We opted to not include LUSER-SW examples as the DW variant outperformed it and wanted to avoid
> increasing the number of images unnecessarily.
>
> [1] M. Louboutin, “Modeling for inversion in exploration geophysics,” Ph.D. dissertation, Georgia Institute of
> Technology, 2020.

---

### Official Review · Reviewer_3hmh · 2022-10-26

**Confidence:** 5
**Clarity, Quality, Novelty And Reproducibility:** 1.Clarity/ Quality
**Correctness:** 3
**Technical Novelty And Significance:** 2
**Empirical Novelty And Significance:** 2
**Recommendation:** 5

**Strength And Weaknesses:**

Strengths:

The studied problem is interesting and worth devoting to. The main advantages of the presented technique are the reduced execution time of the data-consistency layer compared to the one based on deep equilibrium models (DEQ4IP) and improved network capacity compared to traditional deep unfolding (LU-DnCNN).

Weakness:

However, there are some issues that I would like to highlight:

1. One major concern is that the proposed method seems only improve the neural networks learning capacity by running it multiple times within each unrolling iteration. When using the same number of parameters, the memory costs suppose to be the same with normal deep unfolding networks. Unlike DEQ, it not fundamentally solves the memory burden issue when unfolding more iterations (layers) or using more sophisticated networks such as transformers.

2. The conducted experiments cannot sufficiently support the authors’ claim and the numerical results are somehow vague to read. For example, the various properties of different approach in Table 4 are not well supported by the explicit numerical validation. One better way is to present the training/validation loss against time and epoch, compared to other baseline methods. Besides, it is unclear why DEQ4IP has the same PSNR/SSIM results to LU-DnCNN for MRI reconstruction. Since DEQ4IP can run more iterations during training, it should result in better performance, as reported in [Gilton et al. (2021)].

3. While the proposed method aims to improve computational efficiency of the forward model, no large-scale imaging tasks are conducted in this paper, which also diminishes the contribution.


**Summary Of The Paper:**

The proposed work, LUSER, aims to improve the computational efficiency of deep unfolding network at training for solving  general image inverse problems.  To do so,  the proposed LUSER incorporated the fixed-point training of the neural network itself within each unrolling iterations by using the deep equilibrium models (DEQ). DEQ is a recently proposed framework for memory efficient learning of an infinite-depth unrolled network as implicitly defined by a fixed point of an operator. The authors have also conducted experiments to show that their approach is able to get similar quality reconstructions with reduced memory, compared to normal deep unfolding.

**Summary Of The Review:**

Consequently, given the pros and cons on balance, I feel this is a very borderline paper, and I vote for borderline reject tentatively

---

> ### Author Response · Authors · 2022-11-18
> **Response to Reviewer 3hmh Part 1**
>
> We thank Reviewer 3hmh for their thoughtful review. We respond to their highlighted issues below.
> 1. It’s important to clarify the usage of memory under the two contexts talked about in the paper. The first
> relates to the storage memory of the network and is closely tied to the number of parameters. Although
> this plays a role in memory usage during training, it is not the main contributor. The second reference of
> memory corresponds to memory used during training (typically the RAM of a GPU). We make this distinction
> because although our proposed models may have a similar number of parameters/storage memory, it does use
> significantly less training memory, which is the bigger bottleneck in training such models.
> However, we acknowledge that the reviewer makes an excellent point that LUSER does not fundamentally
> solve training memory concerns when attempting to increase the total number of unrolled iterations when
> compared to DEQ4IP. However we would like to bring the following points to the attention of the reviewer.
>     * It is worth noting that most state-of-the-art models (such as those found in the fastmri challenge [1]) still
> stay within the Loop Unrolled framework, suggesting that in practice the framework still holds merit
> despite the limitation.
>     * Figure 2. found in [2] empirically shows a diminishing return for increasing the max iterations (during
> training) of a loop unrolled network. More specifically, the max performance achieved did not occur with
> the longest network (trained for 14 unrolled iterations), but rather at a shorter network, so in practice
> these networks do not need to extend to the same number of iterations of DEQ4IP and may in fact lead to
> degraded performance.
>     * Although not present in the experiments outlined in the paper, we believe DEQ4IP has its own challenges
> when attempting to scale for certain problems. In particular, when the forward model’s computation
> becomes non-negligible relative to the convolutional layers, running many iterations can be a drawback
> rather than a benefit. The slowest model in our experiments was CT that took approximately 0.006
> seconds to run doesn’t demonstrate this drawback. In order to demonstrate this, we ran some simulated
> forward models with run times on the order of 0.05 seconds and 0.5 seconds by introducing an artificial
> delay. Even at 0.05 seconds, the speed benefits of a LU structure, even with the increased run times
> introduced by multiple DEQ regularizers, outperforms the DEQ4IP architecture. There exists plenty of
> forward models with run times even longer than 0.05 seconds, with some Seismic examples hitting 2
> minutes per forward run [3]. Although we have not yet used LUSER to tackle problems on this scale,
> we do believe that there is a regime where LUSER strikes a good balance of memory vs run-time vs
> performance. More details can be found in Section 4.4.
>
>
> The reviewer also mentioned more sophisticated networks as the regularizer. Although not explored in this
> work, we believe that principles outlined in our paper of converting existing architecture into deep equilibrium
> models could also benefit these more sophisticated networks and still achieve comparable memory savings
> when compared to their non equilibrium variants. Although not used for images, [4] demonstrate one such
> conversion of a transformer based network. [4] also show that the equilibrium variants do not lead to a huge
> loss in performance (and in fact sometimes do better) which is empirically backed up by our experiments

---

> > ### Author Response · Authors · 2022-11-18
> > **Response Part 2**
> >
> > 2\. At the reviewers request, we have gone back and ran a hyperparameter sweep over the max number of allowable
> > iterations for DEQ4IP. This has resulted in slightly higher performance across most experiments. However, the
> > curve of performance vs max number of allowable iterations was not monotonically increasing, with peak
> > performance being achieved at different values across the experiments. For example, the numbers reported in
> > the table were accomplished with a max iteration of 40. For the MRI task, increasing to 60 would achieve the
> > peak performance, but for the de-blurring task, increasing to 60 leads to a degradation. As such, we have opted
> > to leave the values in the table as is, and would simply re-emphasize that the numbers reflect similar levels of
> > performance between DEQ4IP vs the proposed model rather than claim that our proposed model outperforms.
> > It is interesting to note that while as the DEQ4IP model for MRI seemed to plateau with increasing max
> > iterations, the same trend did not hold true for CT and CelebA as one would expect.
> >
> > We have also restructured the results section to better explain the various components of Table 4. We have
> > removed training stability, since it is primarily anecdotal rather experimentally and numerically verified. In
> > addition, we clarify what we mean by expressiveness. We explain it as the performance relative to network
> > depth. LUSER is able to achieve comparable performance across the different tasks with a much shallower
> > model. And finally, we expand the section on run times with additional analysis. We add a qualification to the
> > corresponding rows to note under certain conditions, when the forward model begins to dominate calculation
> > (which can occur even at the relatively modest case of 0.05 seconds to run a forward model).
> >
> > 3\. We agree with the reviewer that large scale tasks are of great interest and aim to tackle them in future works.
> > This article is an initial exploration for potential methods that may aid in handling large scale problems.
> >
> > 4\. We have taken the reviewers feedback into account with respect to clarity/quality and have added both the
> > stopping criteria as well as a reference to how anderson acceleration is applied. We have also updated Table 3
> > with the respective input sizes of the 3 tasks.
> >
> > [1] J. Zbontar, F. Knoll, A. Sriram, et al., Fastmri: An open dataset and benchmarks for accelerated mri, 2018. DOI:
> > 10.48550/ARXIV.1811.08839. [Online]. Available: https://arxiv.org/abs/1811.08839.
> > [2] D. Gilton, G. Ongie, and R. Willett, “Deep equilibrium architectures for inverse problems in imaging,” IEEE
> > Transactions on Computational Imaging, vol. 7, pp. 1123–1133, 2021.
> > [3] M. Louboutin, “Modeling for inversion in exploration geophysics,” Ph.D. dissertation, Georgia Institute of
> > Technology, 2020.
> > [4] S. Bai, J. Z. Kolter, and V. Koltun, “Deep equilibrium models,” Advances in Neural Information Processing
> > Systems, vol. 32, 2019

---

### Decision · Program_Chairs · 2023-01-20

**Decision:**

Reject

**Justification For Why Not Higher Score:**

Lack of clarity and poor empirical performance. Reviewers all recommend rejection.

**Justification For Why Not Lower Score:**

N/A

**Metareview: Summary, Strengths And Weaknesses:**

The paper proposes combining a loop unrolled network with deep equilibrium model to solve inverse problems such as CT, MRI, and image deblurring. The paper was found to lack clarity, and the performance of the proposed method did not appear to improve upon deep equilibrium models.